

**Potential of multispectral synergism for observing ozone pollution by combining IASI-NG and UVNS measurements from EPS-SG satellite**

**Lorenzo Costantino [(1)], Juan Cuesta[(1)], Emanuele Emili[(2)], Adriana Coman[(1)], Gilles Foret[(1)], Gaëlle Dufour[(1)], Maxim Eremenko[(1)], Yohann Chailleux[(1)], Matthias Beekmann[(1)], Jean-Marie Flaud[(1)]**

*[(1)] LISA, CNRS UMR7583 Université Paris-Est Créteil et Université Paris Diderot*
*61 Av. Général de Gaulle, 94010, Créteil, France*

*[(2)] CERFACS*
*42 Av. G. Coriolis, 31057, Toulouse, France*

**ABSTRACT**

Present and future satellite observations offer a great potential for monitoring air quality on daily and global basis. However, measurements from currently in orbit satellites do not allow using a single sensor to probe accurately surface concentrations of gaseous pollutants such as tropospheric ozone (Liu et al., 2010). Using single-band approaches based on spaceborne measurements of either thermal infrared radiance (TIR, **Eremenko et al., 2008**) or ultraviolet reflectance (UV, **Liu et al., 2010)** only ozone down to the lower troposphere (3 km) may be observed. A recent multispectral method (referred to as IASI+GOME-2) combining the information of IASI and GOME-2 (both onboard MetOp satellites) spectra, respectively from the TIR and UV, has shown enhanced sensitivity for probing ozone at the lowermost troposphere (LMT, below 3 km of altitude) with maximum sensitivity down to 2.20 km a.s.l. over land, while sensitivity for IASI or GOME-2 only peaks at 3 to 4 km at lowest (**Cuesta et al., 2013**). Future spatial missions will be launched in the upcoming years, such as EPS-SG, carrying new-generation sensors of IASI and GOME-2 (respectively IASI-NG and UVNS) that will enhance the capacity to observe ozone pollution and particularly by synergism of TIR and UV measurements.

In this work we develop a pseudo-observation simulator and evaluate the potential of future EPS-SG satellite observations through IASI-NG+UVNS multispectral method to observer near-surface O3. The pseudo-real state of atmosphere (nature run) is provided by the MOCAGE (MOdèle de Chimie Atmosphérique à Grande Échelle) chemical transport model. Simulations are calibrated by careful comparisons with real data, to ensure the best consistency between pseudo-reality and reality, as well as between the pseudo-observation simulator and existing satellite products. We perform full and accurate forward and inverse radiative transfer calculations for a period of 4 days (8-11 July 2010) over Europe.



In the LMT, there is a remarkable agreement in the geographical distribution of O3 partial columns,
calculated between the surface and 3 km of altitude, between IASI-NG+UVNS pseudo-observations and the
corresponding MOCAGE pseudo-reality. With respect to synthetic IASI+GOME-2 products, IASI-
NG+UVNS shows a higher correlation between pseudo-observations and pseudo-reality, enhanced by about
11%. The bias on high ozone retrieval is reduced and the average accuracy increases by 22%. The sensitivity
to LMT ozone is enhanced on average with 154% (from 0.29 to 0.75, over land) and 208% (from 0.21 to
0.66, over ocean) higher degrees of freedom. The mean height of maximum sensitivity for the LMT peaks at
1.43 km over land and 2.02 km over ocean, respectively 1.03 km and 1.30 km below that of IASI+GOME-2.
IASI-NG+UVNS shows also good retrieval skill in the surface-2km altitude range with a mean DOF (degree
of freedom) of 0.52 (land) and 0.42 (ocean), and an average Hmax (altitude of maximum sensitivity) of 1.29
km (land) and 1.96 km (ocean).
Unique of its kind for retrieving ozone layers of 2-3 km thickness, in the first 2-3 km of the atmosphere,
IASI-NG+UVNS is expected to largely enhance the capacity to observe ozone pollution from space.

## 1 Introduction

The retrieval of tropospheric ozone is a major issue for air quality studies. Ground-level ozone is a priority
air pollutant, causing approximately 22,000 excess deaths per year in Europe (**Amann et al., 2005**). Current
(e.g., MetOp, EOS-Aura, ADEOS) and future (e.g., EPS-SG, MTG-S) satellite observation systems offer a
great potential for monitoring air quality on daily and global basis. Because of their global coverage every
day, they can be used in synergy with regional Chemical Transport Models (CTM) for full data assimilation
(e.g., **Coman et al., 2012**) or inter-validation (e.g., **Zyryanov et al., 2012**). Recent spaceborne instruments
as the Infrared Atmospheric Sounding Interferometer IASI (**Clerbaux et al., 2009**) and the Global Ozone
Monitoring Experiment-2 GOME-2 (**EUMETSAT, 2006**), onboard MetOp satellite, offer a daily global
coverage appropriate for monitoring pollution. Their ground resolution is relatively fine, with four 12 km-
diameter pixels spaced by 25 km (at nadir) for IASI and 80×40 km² ground pixels for GOME-2.
Both TIR and UV observations are able to provide vertical information on ozone concentration. UV sounders
were traditionally used for the stratosphere but recently also for the troposphere (e.g. **Liu et al., 2010**) and
TIR sensors are particularly sensitive to tropospheric ozone, down to the lower troposphere (below 6 km of
altitude, **Eremenko et al., 2008**). However, it has been shown that using spaceborne observations from one
spectral domain (either TIR or UV), only ozone down to 3-4 km of altitude at lowest may be observed (**Foret
et al., 2014**). Recent studies combine the information on ozone distribution from radiance measurements of
different spectral domains. Numerical studies (**Landgraf and Hasekamp, 2007; Worren et al., 2007**)
showed a significant improvement in the sensitivity to retrieve ozone in the lowest 5 km of the troposphere
combining TIR and UV measurements respectively from TES (Tropospheric Emission Spectrometer) and
OMI (Ozone Monitoring Instrument) sounders (both onboard Aura satellite).
More recently, new retrieval approaches have shown the capability to derive ozone profiles from the
multispectral synergism from real satellite measurements at the TIR and UV. **Fu et al. (2013)** combined





measurements from TES (for TIR) and OMI (for UV) sensors and founded a clear improvement in retrieval sensitivity and vertical resolution in the troposphere as well as a sensitivity and accuracy enhancement below 700 hPa, compared with either instrument alone. Cuesta et al. (2013) developed a multispectral synergism of IASI (for TIR) and GOME-2 (for UV) spectra capable of observing for the first time from space ozone

plumes located below 3 km of altitude, defined here as the lowermost troposphere (LMT). The latter approach (referred to as IASI+GOME-2 and used in the following of this paper) decreases the altitude of maximum sensitivity of O3 satellite retrievals, peaking on average at 2.2 km height over land (about 800 m below single-band methods), and enhances the degree of freedom in the LMT by about 40% with respect to single-band retrievals. IASI+GOME-2 uses two radiative transfer codes, KOPRA (Karlsruhe Optimized and

Precise Radiative transfer Algorithm) and VLIDORT (Vector Linearised Discrete Ordinate Radiative Transfer). The inversion algorithm is integrated in the inversion module KOPRAfit. It uses a Tikhonov-Phillips-type altitude dependent regularisation that optimizes sensitivity to LMT (**Eremenko et al., 2008**), maximizing the degree of freedom (DOF) and minimizing the total retrieval error simultaneously.

Incoming satellite missions are expected to carry new-generation instrumentation capable to provide more
accurate observations of tropospheric composition closer to the surface. This is the case of IASI-NG (Infrared Atmospheric Sounding Interferometer-New Generation) and UVNS (Ultraviolet Visible Near-infrared Shortwave-infrared) spectrometers, onboard the future satellite EPS-SG (EUMETSAT Polar System Second Generation) which is expected to be launched in 2022 on a polar-orbit, similarly to MetOp. UVNS instrument is part of the ESA mission Sentinel-5 payload, dedicated to monitoring the composition of the
atmosphere for Copernicus Atmosphere Services.

The advent of these new-generation sensors may allow significant advances in LMT ozone sensing and air quality monitoring. This enhancement needs to be accurately quantified in order to prepare air quality monitoring systems for these future satellite products.


## 2 Purpose and strategy

The objective of this work is to provide a quantitative assessment of the potential of upcoming new-generation space-based observing systems for monitoring ozone pollution. In particular, we investigate the performance of the multispectral synergism of IASI-NG (for TIR emitted spectra) and UVNS (for UV
backscattered radiances) measurements. IASI-NG will have half of IASI radiometric noise and a factor 2 finer spectral resolution (see Table 1). UVNS will have a higher signal-to-noise ratio (SNR) than GOME-2, a much finer horizontal resolution but a factor 2 coarser spectral resolution. To estimate the performances and errors associated to IASI-NG+UVNS, we adapt the existing IASI+GOME-2 retrieval approach to the technical specifications of IASI-NG and UVNS (Table 1). Hereafter, this method will be referred to as IASI-
NG+UVNS.

We set up a Pseudo-Observation Simulator (POS) which is part of an Observing System Simulation Experiment (OSSE). OSSEs are specific type of sensitivity analysis used to quantify the expected added



value of an Observing System (OS). They have been largely used to analyse the gain of future satellite missions on trace gases monitoring (**Edwards et al., 2009; Claeyman et al., 2011; Zoogman et al., 2011**) and are generally composed of three elements: a nature run, a pseudo-observation simulator and an assimilation run. The nature run defines the pseudo-real state of the atmosphere for the experiment (**Masutani et al., 2010**). The pseudo-observation simulator generates the pseudo-retrievals. It calculates and inverts the spectra as it would be done by the observing system, simulating OS performances and errors. The assimilation run consists in the assimilation of pseudo-retrievals into a new model simulation, different and independent from the nature run.

In this work, we only focus on the first two steps of a typical OSSE: the nature run and the simulator of synthetic retrievals, while the assimilation run is left to a further and future research effort. Pseudo-reality is generated by the chemical transport model MOCAGE (MOdèle de Chimie Atmosphérique à Grande Échelle) that simulates physical and chemical processes of atmospheric gasses and aerosol including clouds (e.g., Josse et al. 2004, Marécal et al. 2015). Classical simulations experiments usually provide approximations of the pseudo-observations by means of a predefined parametrization of the averaging kernels (AVK) that describe the retrieval method sensitivity to true vertical profiles of atmospheric species. If this procedure allows to avoid full radiative transfer calculations and costly computational time, approximated averaging kernels with no (or limited) scene-dependence may fail to replicate the variability of the full radiative transfer calculations (**Sellitto et al., 2013b**); complex scene-dependent parametrizations of AVK (**Worden et al., 2013**) may represent a more useful cost-benefits compromise in case of multispectral/multi-instrument observing systems. In order to avoid these approximations, we perform full and accurate forward and inverse radiative transfer calculations, but for a limited 4 days time period.

We pay particular attention to set-up and run the simulation experiment with a high degree of reliability. Indeed, systematic biases in the key parameters of the model control sensitivity lead to unrealistic performances of the satellite product. This is particularly true for parameters such as cloud fraction, surface temperature and temperature profiles. The consistency of pseudo-reality and POS with respect to real data and existing sensor products is analysed and validated in terms of absolute magnitude and spatial variability of a number of atmospheric variables (ozone concentration, skin temperature, temperature profile) and diagnostic parameters (degree of freedom and altitude of maximum sensitivity of the retrieval algorithm).

In the following paragraphs, we present the methodology used to set up the simulation experiment: the nature run, the forward radiative transfer calculations and retrieval scheme of the OS simulator. Then, we carry out a statistical analysis of MOCAGE pseudo-reality with respect to real IASI-GOME-2 measurements and quantify retrieval errors and sensitivity for IASI+GOME-2 and IASI-NG+UVNS. Finally, we compare pseudo-observations of 0-3 km ozone partial columns of the two observing systems.

**3 Nature run and pseudo-observation simulator**



The pseudo-reality is defined by the MOCAGE model (run at CERFACS laboratory), that provides vertical

profiles of atmospheric state and composition variables on hourly basis. It uses 47 sigma-pressure hybrid levels up to 5 hPa (approximately 35 km of altitude) with a vertical resolution that increases from 150 m (lower troposphere) to approximately 1 km (stratosphere) and a horizontal resolution of 0.2×0.2 degrees. The same model configuration that provides operational air quality forecasts over Europe (**Marécal et. al, 2015**) has been used for this study. We simulate LMT ozone pollution events over Europe, from 8 to 11 July 2010.

Real ozone data, from surface network stations (AIRBASE), the Laboratoire d'Aérologie (LA) IASI product (**Barret et al. 2011, Dufour et al., 2005**) and MLS (Microwave Limb Sounder) V4.2 retrievals have been assimilated into MOCAGE simulation to improve the accuracy of the modeled ozone fields at the surface, in the free troposphere and in the stratosphere. The assimilation is performed hourly using a 3D-Var algorithm, described in Jamouillé et al. (2012).

To simulate the multispectral retrievals from IASI+GOME-2, we select the two model grid points that are closest to IASI and GOME-2 ground pixels respectively. For each point, KOPRA (for TIR) and VLIDORT (for UV) radiative transfer codes calculate the spectra, as observed by the satellite sensors, issued from radiation emitted, scattered (only UV) and absorbed by the surface and the atmosphere between 0 and 60 km of altitude. For all profiles, the radiative transfer models have a vertical resolution of 1 km. As the MOCAGE

model top is set at 35 km of altitude, we complete the atmospheric vertical information between 30 and 50 km with data from a global MOCAGE simulation, run at coarser resolution and with model top at 0.1 hPa (**Emili et al., 2014**), and with climatological profiles above 50 km.

Radiometric random noise for IASI and GOME-2 is added to the raw spectra before their ingestion into the retrieval algorithm. For TIR, the nominal noise standard deviation is taken from literature as 20 nW/

$(cm^2cm^{-1}sr)$ (**Eremenko et al., 2008**). Therefore, we added a noise of 13 $nW/(cm^2cm^{-1}sr)$ roughly accounting the reduction of noise due to apodization. For UV, noise is estimated for each wavelength using Muller matrix radiance response elements (**Nowland et al., 2011; Cai et al., 2012**). The signal-to-noise ratio for GOME-2 is equal to 32, for wavelengths between 290 and 306 nm, and equal to 350 between 325 and 340 nm. Spectra are then ingested into the IASI+GOME-2 retrieval algorithm, that assumes no error in co-

localization of TIR and UV measurements: each IASI spectrum (12 km-diameter pixel) is matched with the co-located GOME-2 spectrum (80×40 km²) within a distance of 1 degree, without any averaging. IASI+GOME-2 retrievals are calculated at the IASI ground resolution and processed independently for each IASI pixel.

Partial cloud cover and aerosols are not explicitly modelled in KOPRA, but their effects in the IASI spectra

are partially compensated by offsets for each TIR micro-window (**Eremenko et al., 2008; Dufour et al., 2010**). In the UV spectra calculations, pixels with partial cloud cover are treated as a mixture of clear sky and cloudy scenes according to the independent pixel approximation (**e.g., Cai et al., 2012**). For more details on IASI+GOME-2 multispectral method refer to **Cuesta et al. (2013)**

The IASI-NG+UVNS retrieval method uses the same procedure as IASI+GOME-2, accounting for some

differences in the specifications of the new instruments with respect to the existing ones that are summarized





in Table 1 (for more information, visit https://directory.eoportal.org/web/eoportal/satellite-missions). Note that for UVNS, only the LEO-UV-2 spectral channel is considered as the pixel resolution of the LEO-UV-1 is much coarser.

UVNS will have a higher SNR than GOME-2 but only the half of its spectral resolution. IASI-NG noise and spectral resolution will be approximately the half and a factor 2 finer than those of IASI, respectively. While IASI-NG will have the same spatial resolution as IASI and hence the same footprint (as MetOp and EPS-SG will fly on a similar polar-orbit), UVNS will have a spatial resolution of 7.5 km which is much higher than that of GOME-2. To simulate IASI-NG+UVNS retrievals, we use the closest UVNS measurements with respect to IASING pixel centre.

For better exploiting IASI-NG+UVNS, we have designed a constrain matrix accounting for the capability of the new sensors. As done by Cuesta et al. (2013), we have adjusted the constraints to keep similar errors to IASI+GOME-2 and enhanced sensitivity between the surface and 3 km.

| | IASI (MetOp-B) | IASI-NG (EPS-SG) | GOME-2 (MetOp-B) | UVNS (EPS-SG) LEO-UV-2 channel |
|---|---|---|---|---|
| Radiometric noise | 20 nW/(cm$^2$cm$^{-1}$sr) | 10 nW/(cm$^2$cm$^{-1}$sr) | - | - |
| Spectral resolution | 0.50 cm$^{-1}$ | 0.25 cm$^{-1}$ | 0.22-0.28 nm ($\lambda$=290-306 nm) 0.24-0.30 nm ($\lambda$=325-340 nm) | 0.5 nm ($\lambda$=300-370 nm) |
| SNR | - | - | 32 ($\lambda$=290-306 nm) 350 ($\lambda$=325-340 nm) | 1000 ($\lambda$=300-370 nm) |
| Spatial resolution | 12 km-diameter pixel spaced by 25 km at nadir | 12 km-diameter pixel spaced by 25 km at nadir | 80×40 km² | 7.5×7.5 km² |
| Spectral sampling | 0.25 cm$^{-1}$ | 0.125 cm$^{-1}$ | 0.12 nm | 0.15 nm |

Table 1. Differences in nominal specifications of IASI-NG and UVNS (EPS-SG) with respect to IASI and GOME-2 (MetOp-B). For UVNS, only the LEO-UV-2 spectral channel is considered.

**4 Inversion algorithm and ozone retrieval**

As previously mentioned, the inversion algorithm of IASI+GOME-2 is an altitude dependent Tikhonov-Phillips regularization method for satellite nadir measurements (**Cuesta et al., 2013**). It is integrated in the KOPRAfit module and optimized for lowermost tropospheric ozone observations. The constrain matrix and parameters are optimized to maximize the degrees of freedom and minimize the error in the LMT, retaining a sufficient accuracy in the upper troposphere and in the stratosphere (**Cuesta et al., 2013**). Three different





ozone a priori profiles, derived from climatological values of **McPeter et al. (2007)**, are selected depending on the pseudo-real tropopause height (TH). We use a mid-latitude a priori (30°-60° N) for TH between 10 and 14 km, a tropical a priori (20°-30° N) for higher TH and a polar (60°-90° N) for TH lower than 10 km.

We provide here a quantitative tool to analyse and quantify the error budget associated to each algorithm. If the inversion occurs in an incrementally linear regime, we can consider that the *total error* of a constrained least square fit method is separated into three components (**Rodgers, 2000)**: 1) a *measurement error*, which is random and due to instrumental limitations; 2) a *smoothing error*, which is due the specific retrieval techniques and to the limited sensitivity of the radiance measurements to the vertical distribution of the considered gas; 3) a *systematic error*.

The KOPRAfit module provides the averaging kernel (AVK) matrix of the inversion, describing the vertical sensitivity of retrievals to true profiles. At given altitude (defined by the matrix column), each row shows the fractional height-resolved part of information for the retrieval that comes from the observed spectrum, while the remaining fraction comes from the a priori. The peak value of each AVK column indicates the height of maximum sensitivity to true profiles for the selected altitude. If we integrate the AVK over the rows (i.e., over the different altitudes), we can deduce the height of maximum sensitivity (Hmax) of the retrieval. In the following, we will be interested in the altitude of maximum sensitivity of LMT ozone retrievals, between the surface and 3 km of altitude (Hmax-3km). According to **Rodgers (2000)**, the trace of AVK matrix gives the DOF, a scalar quantity indicating the number of independent pieces of informations within a measurement. For the LMT, the DOF of ozone partial columns is obtained from the trace of AKV up to 3 km (DOF-3km). It is an easy and direct parameter to quantify the vertical sensitivity of each retrieval.

An example of real IASI+GOME-2 retrievals is provided in Figure 1, that shows O3 partial columns (top image) integrated between the surface and 3 km a.s.l. (O3-3km), DOF-3km (middle) and Hmax-3km (bottom) for the 08 July 2010. White gaps indicate a lack of values due to the presence of relatively high cloud cover (>30%), not available satellite data or due invalid retrievals. The DOF-3km map indicates that O3-3km estimates are closer to true profiles over land, where DOF-3km is higher (0.28) than over ocean (0.22), because of a stronger thermal contrast (i.e., the temperature difference between the surface and the lowest atmospheric layer). DOF and Hmax are the two diagnostic parameters that will be used in this work to quantify the sensitivity of a retrieval method.





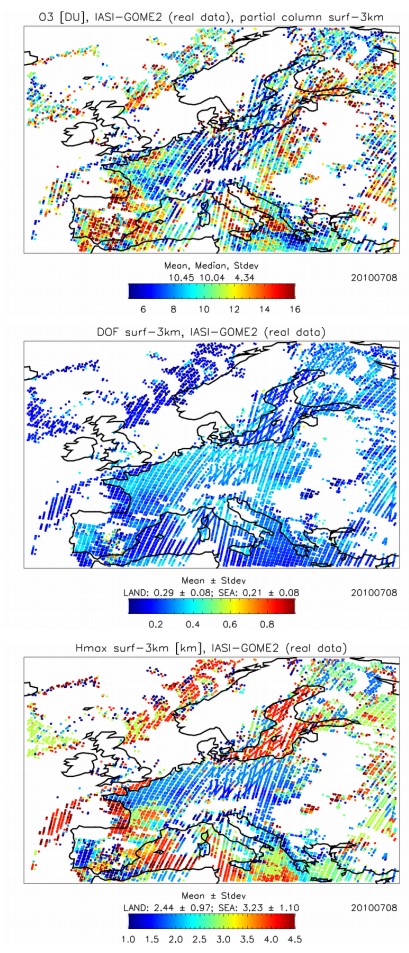

Figure 1. Real IASI+GOME-2 retrievals for the 08 July, 2010. Maps of O3-3km [DU] (top image), DOF-3km (middle), Hmax-3km [km] (bottom). White areas indicate a lack of measurements due to a cloud fractions higher than 0.3 or to invalid retrievals.

The convolution of the pseudo-reality with the AVK matrix (generally referred to as AVK-smoothing) gives an estimate of what retrievals would be without accounting for instrumental limitations (radiometric noise). The *smoothing error* is directly linked to the difference between AVK-smoothed pseudo-reality and pseudo-reality. It can be estimated from the standard deviation of the frequency distribution of this difference. The *measurement error* (i.e. due to radiometric noise) can be estimated from the standard deviation of the difference between pseudo-observations and AVK-smoothed pseudo-reality.





Finally, the difference between pseudo-observations and pseudo-reality (bias) gives an estimates of the inversion algorithm accuracy. Its standard deviation (σ-bias) quantifies the *total error* which is an estimate of the algorithm precision.

**5 MOCAGE pseudo-reality**

In a first analysis, we observed that MOCAGE strongly underestimates the cloud fraction (CLF), that never exceeds a value of 0.3. Therefore, we have estimated an empirical conversion from MOCAGE cloud fraction to more realistic values used as inputs of the pseudo-reality. We then compared the MOCAGE cloud fraction

with GOME-2 apriori values, given from an external algorithm called FRESCO (Koelemeijer et al., 2001), for the whole 8-10 July time period. For those pixels whit CLF < 0.3, we defined two correction factors that minimize the CLF frequency distribution and the mean CLF values, between the two datasets. As the IASI+GOME-2 retrieval algorithm is not sensitive to ozone presence when CLF > 0.3, such cloudy pixels are excluded. The corrected cloud fraction results is calculated as CLF = 20.70×(CLF_mocage$^{1.2}$), where

CLF_mocage represents the original CLF given by MOCAGE. We assume the presence of only low clouds with cloud top pressures above 700 hPa.

As previously mentioned, a more realistic representation of ozone horizontal variability is obtained by assimilation of real ozone data into MOCAGE. To test the reliability of pseudo-real ozone concentration, we compared O3-3km from MOCAGE with real IASI+GOME-2 retrievals. Figure 2 (left) shows the map of

270 model O3-3km data for the 08 July 2010. Light-grey indicates where cloud fraction is larger than 0.3. For consistency with Figure 1, we only show those MOCAGE pixels that would have been selected by IASI+GOME-2 algorithm on that day (i.e., co-located and near-simultaneousness with IASI and GOME-2 footprint at the time of MetOp overpass).

The normalized frequency distributions of Figure 2 (right) show an overall consistence between MOCAGE

outputs and IASI+GOME-2 retrievals, in terms of O3-3km magnitude and spatial distribution. MOCAGE (red line) seems to overestimate the regional mean O3-3km by 2.16 DU (21%). This difference is mostly due to the reduced sensitivity of IASI+GOME2 retrievals to ozone below 3km of altitude. If we account for this effect, smoothing the PR by real AVK (PR*realAVK), the resulting O3-3km distribution (blue line) shows an average value much closer to real data. The positive difference of 0.74 DU (7%) that remains between

PR*realAVK and real IASI+GOME-2 might be linked to systematic explained underestimation of cloud fraction. overestimation of ozone production in MOCAGE model, coming from the systematic underestimation of cloud fraction. A positive bias of MOCAGE at the LMT is also shown by Zyryanov et al. (2012). The larger standard deviation of IASI+GOME-2 and the smaller variability of MOCAGE outputs might be due to the radiometric noise of satellite real data and to the naturally higher variability of real data

than for model simulations.





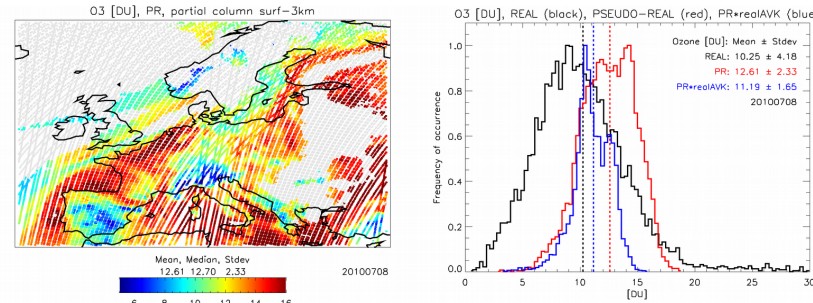

Figure 2. MOCAGE simulation for the 08 July, 2010. (left image) Map of pseudo-real O3-3km [DU]. For consistency with Figure 1, we consider only those MOCAGE pixels that would have been selected by IASI+GOME-2 retrieval algorithm on that day. (right image) O3-3km [DU] normalized frequency distributions of real IASI+GOME-2 measurements (black), pseudo-real MOCAGE data (red) and PR smoothed by real AVK. Dashed lines show mean O3-3km [DU] values of each distribution.

The analysis of pseudo-reality was also performed by comparison of surface temperatures (Ts) from MOCAGE and retrievals, using IASI real spectra within the IASI+GOME-2 algorithm. Figure 3 shows maps of Ts derived from real IASI measurements (top left) and model simulations (top right) for the 08 July 2010, together with the histogram of their normalized frequency distribution over land (bottom left) and over ocean (bottom right). We consider only those pixels with skin temperature in the 5-45 °C range and where the absolute Ts difference between real measurements and MOCAGE is lower than 10 °C. Histograms of Figure 3 clearly indicates that MOCAGE (red) overestimates on average the ocean surface temperature and underestimates land surface temperature with respect to IASI data (black). This is true for the whole 08-10 July time period and the mean difference between real and MOCAGE Ts is equal to -1.42 °C (ocean) and +0.67 °C (land) . Further analysis also shows that MOCAGE overestimates the atmospheric temperature profile in the first 6 km of altitude. For the whole period, the mean difference between real and MOCAGE temperature is equal to -2.69 (0-1km), -1.41 (1-2km), -1.51 (2-3km), -1.14 (3-4km), -0.88 (4-5km) and -0.51 (5-6km) °C. These values have been used to modify MOCAGE temperatures before running radiative calculations and inversion algorithms, in order to consider realistic thermal contrasts for pseudo-reality. It should be mentioned that this work of nature run calibration against real observations and especially for meteorological variables (important for the retrievals) is mandatory to get a realistic pseudo-reality. We believe that it is one important strength of this work to have taken care of these aspects that are often not really tackle in OSSE studies.





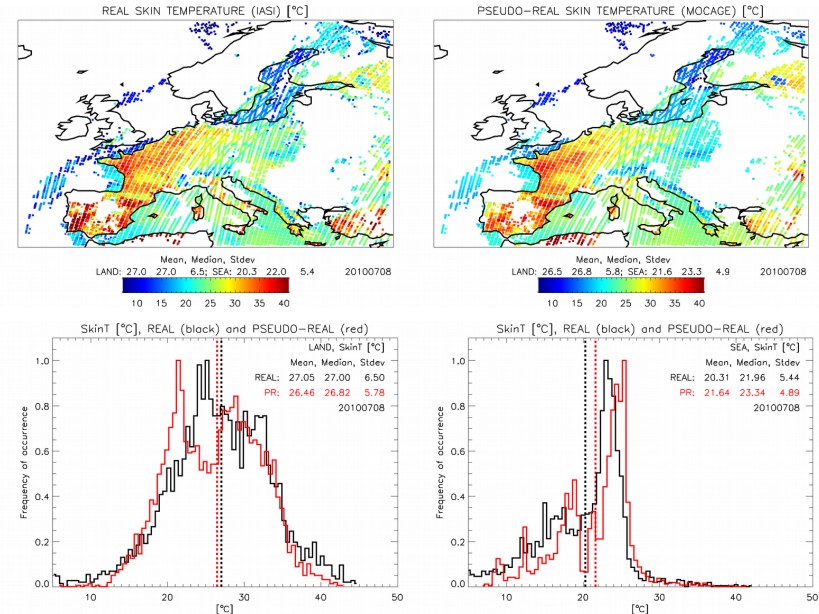

Figure 3. (top) maps of Ts, from real IASI measurements (left) and MOCAGE simulation (right) before correction, for the 08 July 2010. For consistency with IASI data, we consider only those MOCAGE pixels that would have been selected by IASI+GOME-2 retrieval algorithm on that day. (bottom) histograms of Ts normalized frequency distributions for real data (IASI, black line) and pseudo-real outputs (MOCAGE, red line), over land (left) and ocean (right). Dashed lines indicate the mean value. We consider only those pixels with skin temperature in the 5-45 °C range and where the absolute Ts difference between real measurements and MOCAGE is lower than 10 °C.

## 6 Results

After the correction of MOCAGE cloud fraction and temperature fields toward more realistic values, we have used MOCAGE atmospheric profiles to simulated TIR radiances and UV reflectances by KOPRA and VLIDORT respectvely. We than added radiometric random noise to raw spectra and then processed the simulated spectra by IASI+GOME-2 and IASI-NG+UVNS inversion algorithms. Retrievals are provided at the IASI ground resolution. Statistics of retrieval sensitivity and LMT ozone partial columns are presented in the following paragraph.

### 6.1 Retrieval sensitivity





As previously mentioned, multispectral retrieval sensitivity is quantified in terms of lowermost tropospheric
DOF and Hmax. The first row of Figure 4 shows DOF-3km maps from IASI+GOME-2 (left) and IASI-
NG+UVNS (right). To test the general consistency of retrieval performances between the POS and reality,
pseudo-real DOF-3km and Hmax-3km from IASI+GOME-2 (Figure 4) are compared with real satellite
products (Figure 1). We already observed that the MOCAGE geographical distribution of O3-3km is realistic
over Europe (Figure 1 and Figure 2). Figure 4 also shows that pseudo-real and real retrievals from
IASI+GOME-2 are consistent in terms DOF-3km (first row) and Hmax-3km (second row) for both
magnitude and spatial variability. On average, the regional DOF-3km is equal to 0.29±0.07 (over land) and
0.21±0.05 (over ocean) for the synthetic retrievals and to 0.29±0.08 (over land) and 0.21±0.08 (over ocean)
for the real case. The peak of maximum sensitivity in the LMT is equal to 2.49±0.62 km (over land) and
3.40±0.68 (over ocean) for the synthetic retrievals and to 2.44±0.97 km (over land) and 3.23±1.10 km (over
ocean) for the real case. Results are consistent (within statistical uncertainties) with real IASI+GOME-2
sensitivity values obtained by Cuesta et al. (2013) over the same region for the 19-20 August 2009. They find
a DOF-3km of 0.34±0.04 (land) and 0.23±0.04 (ocean), and a Hmax-3km of 2.20±0.50 (land) and 3.42±0.59
(ocean).

The IASI-NG+UVNS map of DOF-3km (right) shows an average increase in the degree of freedom at 3km
over both land and ocean of about 150% and 200%, respectively with respect to IASI+GOME-2, with mean
values that grow from 0.29 (land) and 0.21 (ocean) with IASI+GOME-2 to 0.72 (land) and 0.64 (ocean) with
IASI-NG+UVNS. Note that IASI-NG+UVNS uses an optimised constraint to enhance sensitivity in the
LMT. Accordingly, with the new spectral method the peak of LMT sensitivity to ozone (Hmax-3km, second
row of Figure 4) decreases on average by about 1.1 km over land (from 2.50 to 1.41 km) and 1.27 km over
ocean (from 3.40 to 2.13 km). Sensitivity differences between land and ocean are strong for IASI+GOME-2
and are also present in IASI-NG+UVNS retrievals, because of the different thermal contrast which is greater
over land. The third row of Figure 4 shows that normalized frequency distributions of DOF-3km are wider
and shifted to higher values when using new-generation sensors (right image), with a similar mean gap of 0.8
as for IASI+GOME-2 between land (red line) and ocean (blue line). It is remarkable that in several cases
IASI-NG+UVNS method shows a DOF-3km ≥ 0.8 (22% of total) and a Hmax-3km ≤ 1.5 km (37% of
total).

The fourth and fifth rows of Figure 4 report two examples of AVK vertical profiles at retrieval altitudes of 0,
1, 2, 3 km (red lines) and 4, 5, 6 km (blue lines) and 7, 8, 9, 10, 11, 12 km (black lines) for two specific
pixels over land and ocean (black-yellow spot in maps) with thermal contrast (reported in figure) particularly
high and equal to 14.5°C (land) and 5.9°C (ocean). Stratospheric DOF (surface-60 km) is consistently
increased with IASI-NG+UVNS, by 1.59 (+31%) over land and 1.40 (+29%) over ocean, and tropospheric
DOF (surface-12km) is increased by 1.19 (+75%) over land and 1.09 (+76%) over ocean. However, the
maximum gain is obtained in the LMT where DOF-3km increases by 0.61 (+179%) over land and 0.58
(+305%) over ocean. The dotted line indicates the height of maximum sensitivity in the LMT. As expected,
IASI+GOME-2 shows better retrieval performances over land (Hmax-3km = 2 km) than over ocean (Hmax-

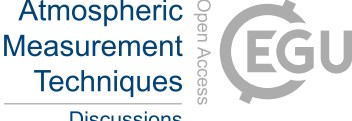



3km = 4 km). The use of IASI-NG+UVNS further decreases Hmax-3km over land and even more over ocean, down to 1 km of altitude in both cases.


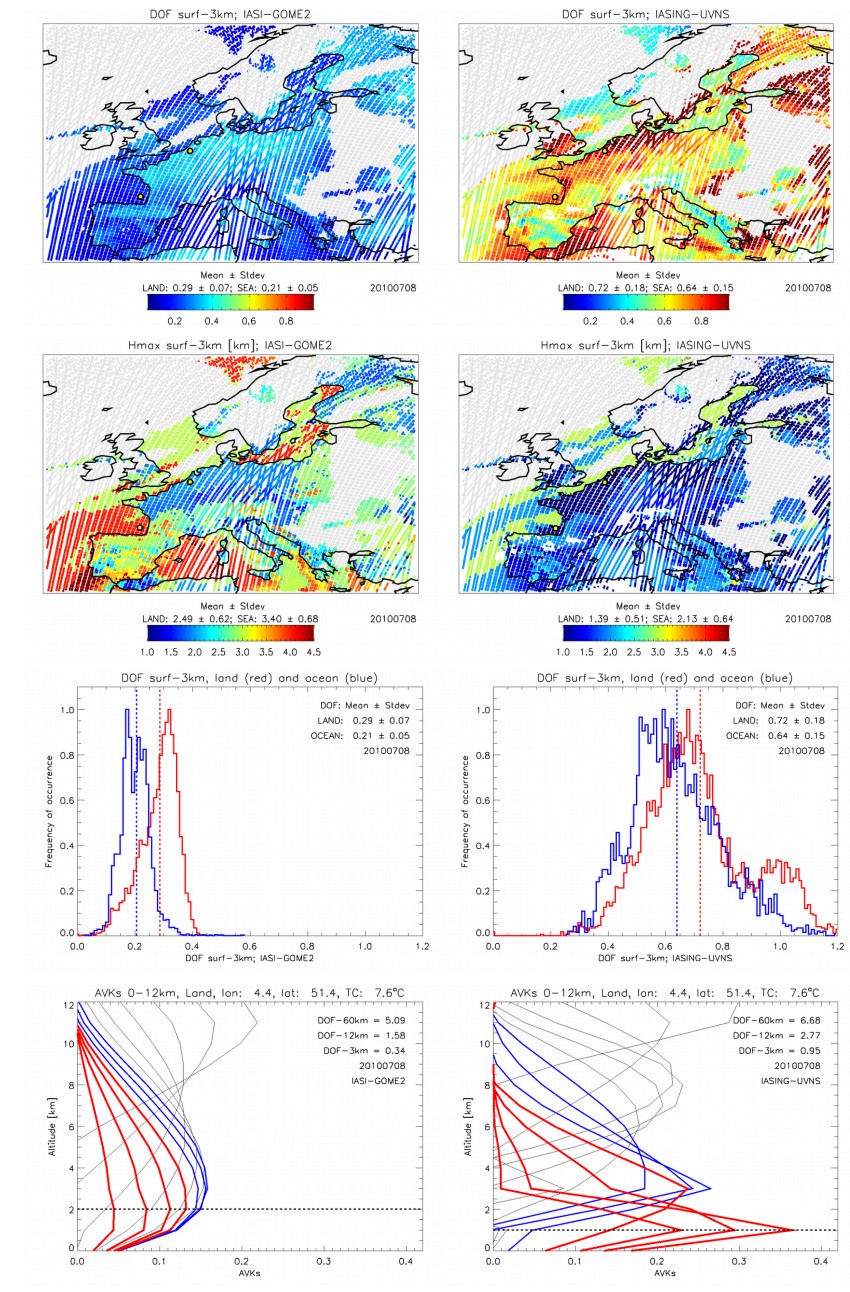





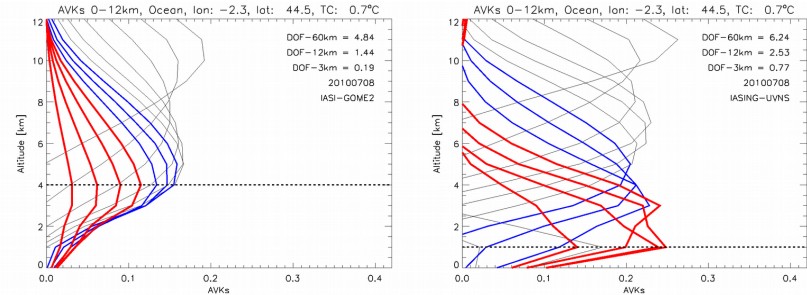


Figure 4. IASI+GOME-2 (left column) and IASI-NG+UVNS (right column) data, for the 08 July 2010. (first row) maps of DOF-3km pseudo-observations (PO), grey colour indicates where CLF is larger than 0.3. (second row) maps of Hmax-3km from PO. (third row) histograms of DOF-3km normalized frequency distribution from PO, over land (red) and ocean (blue). (fourth row) AKV vertical profiles at retrieval

altitudes of 0, 1, 2, 3 km (red lines) and 4, 5, 6 km (blue lines) and 7, 8, 9, 10, 11, 12 km (black lines) for a pixel over land (black-yellow spot in the map) in a highly polluted area. Values of stratospheric DOF (0-60 km, referred to as "total") and DOF-3km are reported in figure. Dashed line indicates the altitude of maximum sensitivity, in the LMT. (fifth row) same as before, but over ocean.

For the whole time period (8-11 July), averaged values of regional mean DOF-3km and Hmax-3km for

IASI+GOME-2 and IASI-NG+UVNS are presented in Table 2, with real IASI+GOME-2 data reported in parenthesis next to pseudo-real ones. The IASI-NG+UVNS retrieval algorithm parametrization has been optimized to increase at most the sensitivity in the lowest layers of the atmosphere (as close as possible to human biosphere), rather than decreasing the total retrieval error. Hence, the major gain of using the multispectral synergism of EPS-SG sensors concerns DOF-3km and Hmax-3km. DOF-3km is more than

doubled and tripled over land and ocean, while Hmax-3km decreases by 1.03 km and 1.3 km respectively.

**6.2 LMT ozone retrievals**

Figure 5 shows maps of O3-3km pseudo-observations (first row) and AVK-smoothed pseudo-reality (second

row), for IASI+GOME-2 (left) and IASI-NG+UVNS (right). In terms of absolute ozone concentrations, both IASI+GOME-2 and IASI-NG+UVNS have an overall good agreement with the MOCAGE pseudo-reqlity. The two multispectral methods differ by less than 3% in estimating the mean O3-3km value over the whole region. However IASI-NG+UVNS shows a larger spatial variability and capture more efficiently some high O3-3km values off the coat of Northern Spain, France, Holland, and Mediterranean Basin. This is even more

visible looking at the AVK-smoothed pseudo-reality (Figure 5, second row), where radiometric noise is not present. O3-km patterns smoothed by IASI-NG+UVNS averaging kernels are closer to MOCAGE product (Figure 2) because of the stronger sensitivity to true profiles.




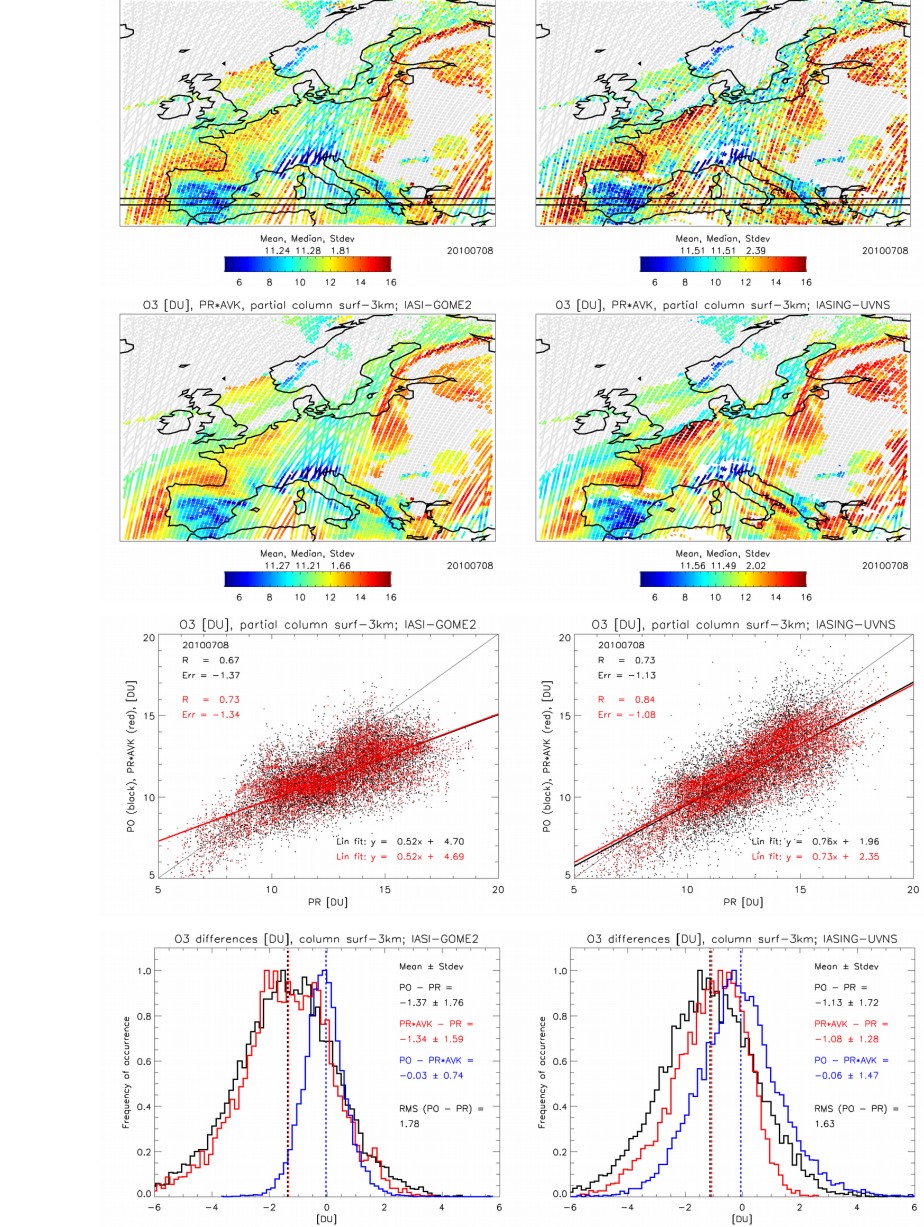

Figure 5. IASI+GOME-2 (left column) and IASI-NG+UVNS (right column) data, for the 08 July 2010. (first row) maps of 03-3km pseudo-observations (PO), grey colour indicates where CLF is larger than 0.3. Black lines indicate the [38N, 39N] latitude band. (second row) maps of O3-3km from AVK-smoothed pseudo-reality (PR*AVK). (third row) scatterplot of O3-3km [DU] from pseudo-observations (black, y-axis) and



AVK-smoothed pseudo-reality (red, y-axis) versus pseudo-reality (PR, x-axis). Linear correlation coefficient,

mean error and linear fit equation are reported in figure. (fourth row) histograms of normalized frequency distribution of O3-3km differences [DU] between PO and PR (black), PR*AVK and PR (red), PO and PR*AVK (blue). Mean value and standard deviation of each difference is reported in figure, together with PO-PR root mean square (RMS).


As a consequence, the correlation between pseudo-retrievals and pseudo-reality is enhanced for IASI-NG+UVNS. Figure 5 (third row) presents the scatterplot of O3-3km pseudo-observations (black) and AVK-smoothed pseudo-reality (red), plotted in function of pseudo-reality. With respect to existing instruments (left image), the product of new-generation sensors (right image) are better correlated to MOCAGE data. The

correlation coefficient increases from 0.67 to 0.73 (+9%) and the linear fit slope (reported in figure) of the dispersion plot increases from 0.52 to 0.76 (+46%), sign that high O3-3km values (larger than ~8.5 DU) are less underestimated.

The fourth row of Figure 5 shows the normalized frequency distributions of (pixel-by-pixel) difference in O3-3km between: pseudo-observations and pseudo-reality (PO-PR, black), AVK-smoothed pseudo-reality and pseudo-reality (PR*AVK-PR, red), pseudo-observation and AVK-smoothed pseudo-reality (PO-

PR*AVK, blue). With respect to IASI+GOME-2, the IASI-NG+UVNS mean value of PO-PR distribution (retrieval mean bias) decreases from -1.37 to -1.13 (-17%) and the root mean square (RMS) from 1.78 to 1.63 (-8%). The standard deviation of the bias slightly decreases from 1.76 to 1.72 (-2%). Results indicate a clear and global gain in retrieval accuracy (from bias and RMS) when using EPS-SG sensor instead of

MetOp instrumentation, while retrieval precision (from σ-bias) is not sensibly improved. This is consistent with the design of the constrain matrix of IASI-NG+UVNS. For the whole time period (8-11 July), Table 2 reports regional means of the linear correlation coefficient R, between pseudo-observations and pseudo-real data of O3-3km, as well as the bias, the σ-bias and the RMS of the PO-PR distribution, for both IASI+GOME-2 and IASI-NG+UVNS.


| | DOF-3km | | Hmax-3km [km] | | R | Bias [DU] | σ-bias [DU] | RMS [DU] |
|---|---|---|---|---|---|---|---|---|
| | land | ocean | land | ocean | | | | |
| | | | | | | | | |
| IASI+GOME-2 | 0.29±0.06 | 0.21±0.06 | 2.46±0.60 | 3.39±0.68 | 0.65 | -1.30 | 1.79 | 1.77 |
| (real) | (0.29±0.08) | (0.21±0.08) | (2.51±0.88) | (3.32±1.07) | | | | |
| IASI-NG+UVNS | 0.75±0.19 | 0.66±0.15 | 1.43±0.50 | 2.09±0.60 | 0.73 | -1.01 | 1.70 | 1.55 |
| Gain | +154% | +208% | -1.03 | -1.30 | +11% | -22% | -5% | -0.14% |

Table 2. Averaged values (8-11 July 2010) of regional mean DOF-3km, Hmax-3km [km], R, bias [DU], σ-bias [DU] and RMS [DU] of the O3-3km PO-PR distribution over Europe, for IASI-GOME-2 and IASI-





NG+UVNS. Real IASI+GOME-2 values of DOF-3km and Hmax-3km are reported in parenthesis next to
pseudo-real ones.

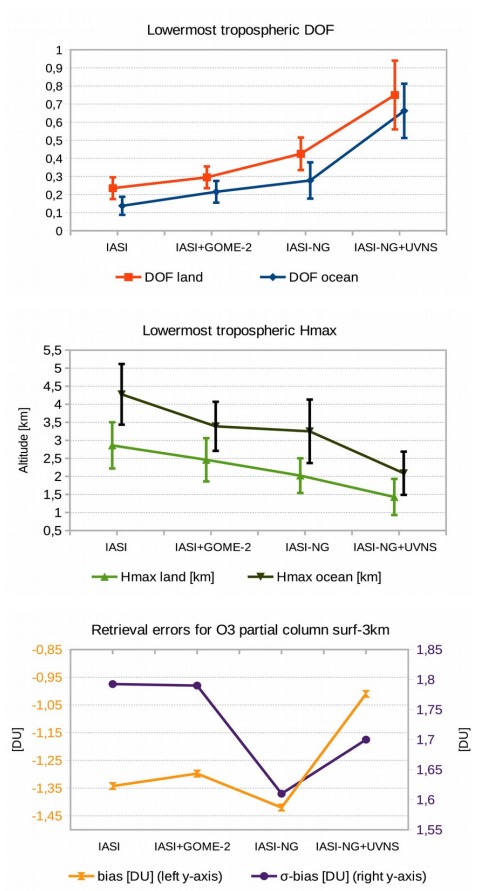

Figure 6. Average values of DOF-3km (top), Hmax-3km (middle), bias and σ-bias (bottom) of the O3-3km
PO-PR distribution over Europe for IASI, IASI+GOME-2, IASI-NG and IASI-NG +UVNS, for the whole 8-
11 July time period over Europe.

Figure 6 shows graphically the mean values of DOF-3km (middle), Hmax-3km (middle), bias and σ-bias
(bottom) of the O3-3km PO-PR distribution for IASI+GOME-2 and IASI-NG+UVNS (Table 2) together
with these same values for IASI and IASI-NG alone, simulated over the same area and time period. Note in
case of IASI alone, the POS seems to overestimate the retrieval performances with a Hmax-3km over land
equal to 2.86±0.64 km, which is only 400 m above that of IASI+GOME-2 instead of the 800 m expected by





Cuesta et al. (2013). It is evident, however, that IASI+GOME-2 represents a clear improvement to IASI alone in terms of DOF-3km, Hmax-3km and bias. At the same time, the technical advances of IASI-NG

allow retrieval performances even higher than IASI+GOME-2 for what concerns DOF-3km, Hmax-3km and σ-bias. On the other hand, when IASI-NG is coupled with UVNS, the IASI-NG+UVNS synergism overpasses by far the retrieval skills of all other configurations in terms of DOF-3km, Hmax-3km and bias.

The high quality retrieval skills of IASI-NG+UVNS in the LMT suggest to go deeper in the lowermost troposphere and investigate ozone sensitivity in the surface-2km layer. Figure 7 shows the DOF-2km (top)

and the Hmax-2km (bottom) for the 8 July 2010, for IASI+GOME-2 (left) and IASI-NG+UVNS (right). With respect to LMT, the retrieval performances of IASI+GOME-2 degradate sensibly in terms of DOF (0.16±0.05 over land 0.10±0,03 over ocean), as the Hmax-2km (2.41±0.63 km over land and 3.34±0.68 km over ocean) remains above 2 km of latitude. On the contrary, IASI-NG+UVNS shows still a relatively high DOF-2km of 0.49±0.17 (land) and 0.40±0.15 (ocean) and a Hmax-2km of 1.25±0.48 km (land) and 2.0±0.67

km (ocean), even better than IASI+GOME2 for the surface-3km partial column.

Averaged over the whole time period, the IASI-NG+UVNS regional mean DOF-2km is equal to 0.52±0.17 (land) and 0.42±0.15 (ocean) with a Hmax-2km of 1.29±0.49 km (land) and 1.96±0.63 km (ocean). As a consequence, the map of O3-2km pseudo-retrievals from IASI-NG+UVNS (Figure 8) is much closer to reality than for IASI+GOME-2. In particular, high ozone values are less underestimated over Holland and

the Mediterranean basin (especially off the coast of East Spain and Southern Italy), where low level ozone layers are present.

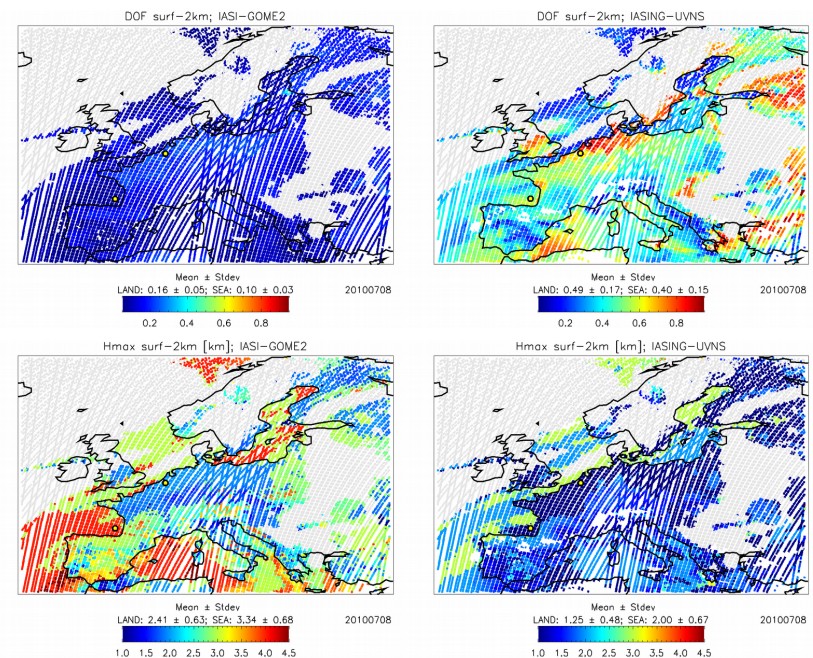






Figure 7. IASI+GOME-2 (left) and IASI-NG+UVNS (right) data, for the 08 July 2010. Maps of DOF-2km (top) and Hmax-2km (bottom) pseudo-observations.


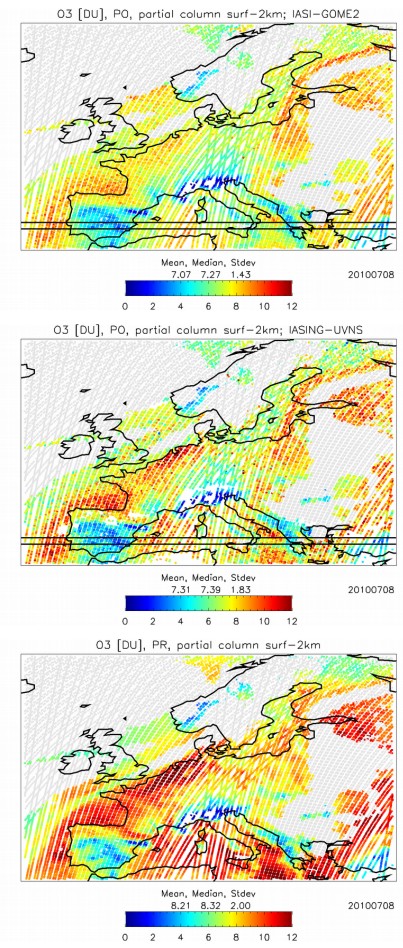

Figure 8. IASI+GOME-2 (top) and IASI-NG+UVNS (middle) pseudo observations of 03-2km and PR (bottom) for the 08 July 2010.

## 6.3 Capacity of resolving layers

Along with sensitivity to surface ozone concentrations, vertical resolution of ozone retrievals is the major limitations of such products. Indeed, the mixing of vertical information screen out the origin of ozone and limits for example their capability to improve models when assimilated.





Figure 9 shows a transect of ozone profiles which allows one to compare the capability of each retrieval to resolve ozone layers. It presents the vertical cross-sections of ozone concentration as observed by
IASI+GOME-2 (top) and IASI-NG+UVNS (middle), with the corresponding MOCAGE output (bottom). Data are averaged horizontally with a resolution of 1x1 degrees, along the latitude band [38°N, 39°N]. Both retrieval methods show an overall agreement with pseudo-reality for ozone plumes with concentration higher than 1.0 mol/µm³.


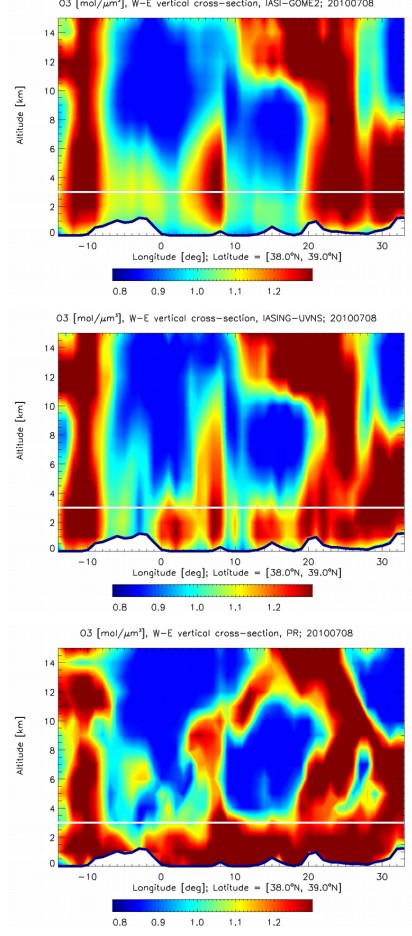

Figure 9. Cross section of ozone concentration [mol/µm³] in the latitude band [38°N, 39°N] (as shown in Figure 5, first row), from the surface to 15 km of altitude, for the 08 July 2010. Data are averaged horizontally with a resolution of 1×1 degrees. (top) IASI+GOME-2 retrievals. (middle) IASI-NG+UVNS retrievals. (bottom) MOCAGE pseudo-reality. White line indicates an altitude of 3 km.




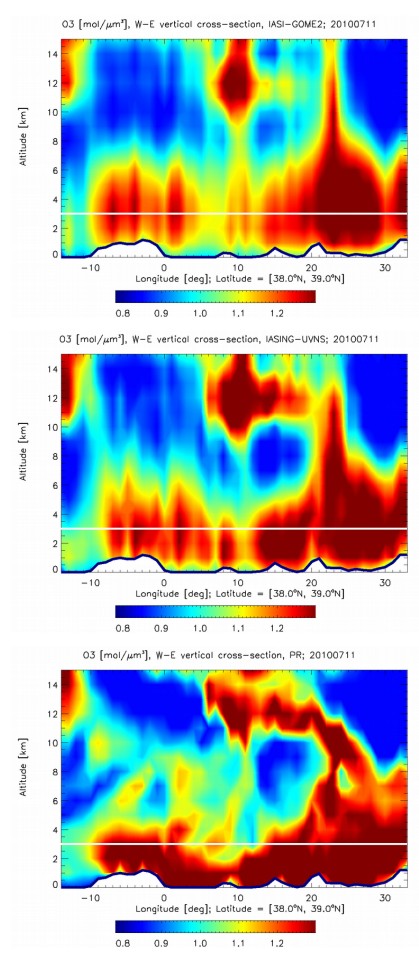

Figure 10. Same as Figure 9, but for the 11 July 2010.

The comparison of IASI-GOME-2 cross-section with the pseudo-real O3 concentration confirms that IASI-
GOME-2 is able to resolve complex vertical ozone distribution in the lowermost and lower troposphere. This
is true for the three ozone peaks (red colour) at [14°W, 10°W], [4°E, 8°E] and [20°E, 30°E] or the moderate
O3 concentrations (yellow-green colour) at [12°W, 5°E] and [9E, 17°E] located below a clean atmosphere.
The vertical structures described by IASI+GOME-2 matches well MOCAGE simulations and are consistent
with real IASI+GOME-2 product, which is able to observe ozone plumes below 3 km, even if it cannot
distinguish whether the ozone plumes are located in the LMT or between 3 and 6km of altitude (Cuesta et al.,
545    2013).




IASI-NG+UVNS shows a finer resolution than IASI+GOME-2, resolving ozone layers of 2-3 km thickness below 3 km of altitude. At [0°E, 5°E] and [8°E, 20°E], ozone concentrations higher than 1.2 mol/μm³ are entirely located in the lowermost troposphere. Even if partially, the scene is captured by IASI-NG+UVNS that is able to depict the signature of LMT ozone, while IASI+GOME-2 fails to detect strong O3
concentrations.

Same as Figure 9, but for the 11 July 2010, Figure 10 shows the O3-3km vertical cross-section in the [38°N, 39°N] latitude band. Again, IASI-NG+UVNS provides more reliable retrievals where ozone concentrations are larger than 1.2 mol/μm³, as at [3°E, 10°E] below 4 km (and in particular the peak at [8°E, 10°E] below 2 km) and at [14°E,19°E] below 3 km. At higher altitudes between 11 and 13 km a.s.l., the 2 km thick ozone
layer at [12°E, 20°E] is well resolved by IASI-NG+UVNS but invisible to IASI+GOME-2.

The ability of IASI-NG+UVNS to identify ozone gradient between the 3-6 km and surface-3km analysed in Figure 11, showing maps of pixel-by-pixel difference in ozone partial columns between the two altitude bands. Negative values (in purple) indicate where the O3 concentration is higher in the LMT. The regional root mean square of the differences is reported in figure, above the colour scale. IASI-NG+UVNS (middle)
clearly shows a better agreement with pseudo-reality (bottom) than IASI+GOME-2 (top), especially over Northern France, Holland and Mediterranean basin.

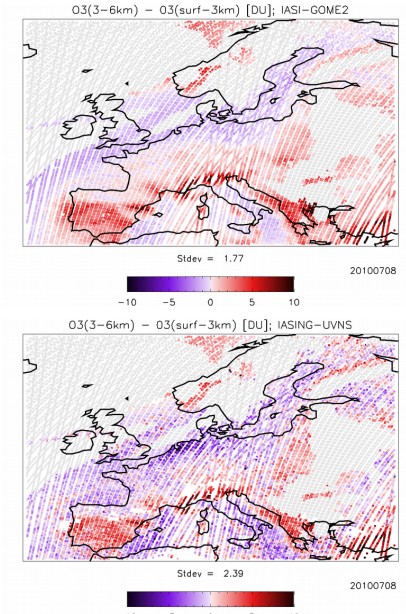






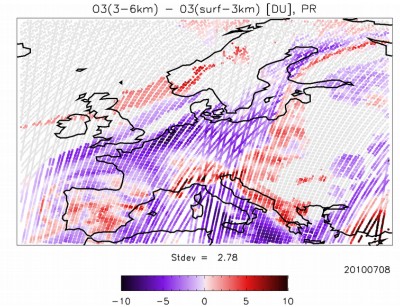

Figure 11. Maps of pixel-by-pixel difference between ozone partial columns [DU] calculated in the surface-3km and 3-6km altitude bands, for the 08 July 2010. Negative values (purple) indicate stronger ozone concentration in the LMT than above. Regional root mean square (RMS) value is reported in figure.

The same dataset shown in Figure 11 is analysed in Figure 12 as a scatterplot. On the x-axis we present pseudo-reality, on the y-axis the pseudo-observations (black) and the AVK-smoothed pseudo-reality (red). Data from IASI+GOME-2 (left, y-axis) and IASI-NG+UVNS (right, y-axis) are compared with pseudo-reality (x-axis). Both methods seem to identify correctly cases where pseudo-real differences are positive, but IASI-NG+UVNS allows to detect negative values (higher ozone concentration in the LMT than above it) down to -8 DU, while IASI+GOME-2 is limited to -2 DU. In case of EPS-SG sensors, the mean error between pseudo-reality and synthetic retrievals is sensibly less biased by -36%.

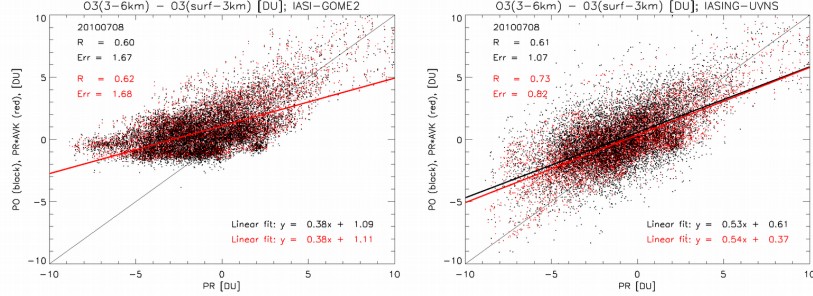

Figure 12. Same data than in Figure 11. (left) IASI+GOME-2 retrievals (black, y-axis), AVK smoothing of pseudo-reality (red, y-axis) and pseudo its pseudo-reality (x-axis). (right) IASI-NG+UVNS retrievals (black, y-axis), AVK smoothing of pseudo-reality (red, y-axis) and pseudo-reality (x-axis). Linear correlation coefficient, mean error and linear fit equation are reported in figure.





**7 Summary and conclusion**

In this work, we quantify the potential of the synergism for LMT ozone retrieval of combining TIR and UV measurements from the new-generation sensors IASI-NG and UVNS (onboard EPS-SG satellite) with respect to the existing IASI+GOME-2 multispectral method. To achieve this goal, we develop a pseudo-

observation simulator, where the nature run is defined by output from the MOCAGE model, where real ozone data from surface network stations, IASI and MLS instruments have been assimilated for a realistic representation of ozone horizontal variability at the surface and the free troposphere. To ensure the highest degree of reliability with respect to the experiment, the pseudo-real atmosphere has been carefully calibrated by a comparison of real data with the POS of IASI+GOME-2. Cloud fraction, skin temperature and

temperature profile were empirically corrected in order to obtain realistic sensitivity of the satellite products. This calibration stage of meteorological variables important for retrievals appears as a key point of the methodology and we believe it should systematically considered in OSSE. Atmospheric and surface spectra are simulated by KOPRA and VLIDORT radiative transfer codes, performing full and accurate forward and inverse radiative transfer calculations. We analyse and compare pseudo-observations of IASI+GOME-2 and

IASI-NG+UVNS from 8 to 11 July 2010 over Europe. Data assimilation analysis in a different chemical transport model independent from MOCAGE is left to future research work.

Over the whole time period, IASI-NG+UVNS estimates of ozone partial columns, calculated between the surface and 3 km of altitude, are highly correlated to the MOCAGE outputs. With respect to IASI+GOME-2, using new-generation sensors the correlation coefficient between O3-3km pseudo-observations and pseudo-

reality increases on average by about 11%, from 0.65 to 0.73, and the retrieval of high ozone values is less underestimated. As a consequence, the retrieval bias is significantly reduced by -22%. In addition, σ-bias slightly decreases by -5%. The bias between PO and PR gives an estimate of the retrieval method accuracy and is strongly improved mostly because of a higher retrieval sensitivity to true profiles in the lower tropospheric layers (below 2-3km). The σ-bias quantifies the *total error* of the multispectral retrieval and

measures the inversion algorithm precision that remains almost constant when migrating from the existing to the new-generation sensor synergism. This is consistent with the fact that we have designed a specific constrain matrix for IASI-NG+UVNS with the purpose to enhance sensitivity in the lower layers.

With respect to IASI+GOME-2, the main gain of IASI-NG+UVNS relies a on mean DOF-3km increase over both land (from 0.29 to 0.75) and ocean (from 0.21 to 0.66), which is respectively 154% and 208% higher

than using present instrumentation. Accordingly, the mean height of maximum sensitivity in the LMT decreases down to 1.43 km over land and 2.09 km over ocean, which is approximately 1.03 km and 1.30 km below IASI+GOME-2. In addition, IASI-NG+UVNS can also provide reliable ozone retrievals below 2 km of altitude, with an average DOF-2km of 0.52 (land) and 0.42 (ocean) and a mean Hmax-2km of 1.29 km (land) and 1.96 km (ocean). It seems to be able to observe ozone layers of 2-3 km thickness and distinguish

if ozone plumes are located in the lowermost troposphere or just above it, between 3 and 6 km.



This unique capability of IASI-NG+UVNS to provide high confidence O3 retrievals in the first 2-3 km of the atmosphere should significantly improve regional ozone estimates for air quality studies. Further work should assess the potential impact on ozone forecasts, when assimilating this new multispectral product in a chemical transport model independent from MOCAGE.

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
