# Peer review of "Potential of multispectral synergism for observing ozone pollution by combining IASI-NG and UVNS measurements from EPS-SG satellite"

_Atmospheric Measurement Techniques, 2016_

## Referee Comment (RC1) · Anonymous Referee #1 · 20 Jan 2017

Please see the review comments in the Supplement pdf file. Thanks.

Please also note the supplement to this comment:
http://www.atmos-meas-tech-discuss.net/amt-2016-374/amt-2016-374-RC1-supplement.pdf

---

## Referee Comment (RC2) · Anonymous Referee #2 · 24 Jan 2017

General Comments:

This paper demonstrates the potential of combining future thermal infrared (IASI-NG) and UV (UVNS) measurements from the EPS-SG satellite to significantly improve retrievals of lowermost tropospheric ozone over existing capability of combining IASI and GOME-2. It uses a pseudo-observation simulator, generating synthetic data based on instrument specifications and then performing inversions from these synthetic data. To clearly show the improvement, the IASI-NG + UVNS retrievals are contrasted with IASI+GOME-2 retrievals in terms of surface-3 km DOF, height of maximum sensitivity to surface-3 km ozone as well as retrieval biases. This study is suitable for publication in AMT. It is very well logically organized. It is also generally well written although the

English usage can be improved and the abstract can be more concise. Some of the figures or tables can be improved and more explanations can be added. Overall, I recommend it to be published after addressing the following comments.

Specific Comments:

1. The abstract seems to be too long with a long introduction. Please make it concise and only highlight the important points.

2. The abstract typically should not have citations.

3. Also, please define acronyms at their first occurrences such as IASI, IASI-NG, GOME-2, EPS-NG, UVNS, a.s.l., in the abstract, and also separately at their first occurrences in the text.

4. L40, L43 and in the text, suggest adding "for signal" after "degree of freedom" as it could be used for signal or for noise.

5. L51, it seems to me that ADEOS is not currently working.

6. L53, this also applies to global CTMs

7. L63, suggest adding "with adequate vertical sensitivity" after "observed" as UV retrievals can be sensitive to ozone down to below 2-3 km although with reduced sensitivity and retrieved at a much broader altitude range as shown in Liu et al. (2010)

8. L65, add "to enhance sensitivity to LMT ozone" after "spectral domains"

9. L74, I would not call this "for the first time" as the study of Fu et al. (2013), published almost at the same time (slightly earlier), also showed similar capabilities (e.g., surface-700 hPa ozone)

10. L120, suggest rephrasing to "For practical consideration of computation, OSSEs typically approximate the pseudo-observations, . . ."

11. L159, the vertical resolution is not a property of radiation transfer models, but the

radiative transfer calculations. I suggest rephrasing it to "All the profiles are sampled to 1-km layers in the radiative transfer calculations"

12. L168-169 and in Table 1, what is the SNR between 306-325 nm?

13. L216-217, the sentence "At given altitude (defined by the matrix column), each row shows " is confusing and should be rephrased, as both column and row refer to a given altitude and the sentence reads like, "For the given column, each row . . ."

14. In section 4, it would be very useful to mention other retrieval variables (some of them are interfering) in addition to ozone even it is mentioned in Cuesta et al. (2013).

15. L263, the sentence is not accurate as the cloud can enhance ozone sensitivity above clouds, suggest rephrasing to "IASI+GOME-2 retrieval sensitivity to LMT ozone is significantly reduced when CLF > 0.3"

16. L279-282, I don't see the need to understand the small mean IASI+GOME-2 and MOCAGE differences as one is based on retrievals (not real data), and the other is based on assimilation of different ozone products, and too many important factors in retrievals, model simulations/assimilations can cause their differences.

17. L363, it is not clear about what you mean by "with a similar gap of 0.8 . . ."

18. Table 1 & 2: captions typically are put above tables. Please check the requirements. Table 2 caption, it is good to describe what is R, $\sigma$-bias, and RMS in a few extra words. The gains for different columns are likely calculated inconsistently, please consistently use the IASI-GOME-2 as the reference point. The gain for the last one should be (1.55-1.77)/1.77=-12.4%

19. In the 3rd row of Figure 5, there are no significant differences in the linear regression slope between black and red. Typically, you want to show how well your pseudo observations can capture the pseudo-reality or pseudo-reality with AVK (typically in validation). I think that it is better to show the PO (y-axis) vs (PR or PR*AVK). The slope for PO vs PR*AVK should be much closer to 1. This is also the case for Figure 12.

Also in the 4th row, the standard deviation between PO and PO*AVK is much larger for the IASI-NG+UVNS (1.47 DU) than that of IASI+GOME2 (0.74), suggesting that the retrieval precision is worse for IASI-NG+UVNS. It is good to mention this and give an explanation.

20. In Figure 6 bottom and L475-479. Can you please explain why sigma-bias for IASI-NG+GOME2 becomes larger than that for IASI-NG alone while the sigma-bias for IASI-NG alone is smaller than for IASI alone?

21. Figure 9, I would like to suggest the same thing as the first reviewer, i.e., plotting the data in more conventional units of ppbv.

22. The synthetic retrievals show the potential of synergism of combining IASI-NG and GOME-2 for enhancing LMT retrievals. However, real retrievals might not be able to realize all the potentials due to mismatch of IASI-NG and UVNS and the retrieval challenges in TIR or in UV or the combination. Take the UV for example, the signal to noise ratio is assumed to be 1000. In practice, it might be difficult to fit the data well enough to this level. Also there can be strong correlation between surface albedo parameters and ozone in the lower troposphere, reducing the potential such enhancement to LMT ozone retrievals. It might be useful to discuss/mention some of the implementation challenges or future work in the conclusion.

Technical comments

1. L21, change "at the lowermost" to "in the lowermost"

2. L23, In abstract, change "spatial missions" to "spaceborne mission"

3. L28, use subscript in O3. It also occurs several times in the text or figure captions

4. L56, change to "the Metop satellites" due to multiple satellites

5. L57, change to "ground resolutions are"

6. L65, change to "Worden"

7. L70, change "at" to "in", or change to "real TIR and UV satellite measurements"

8. L84, L91 and several other occurrences, add "of" after "generation"

9. L88, change to "similar"

10. L167, change to "Nowlan"

11. L182, "LEO-" are not needed before "UV-1" and "UV-2"

12. L185, change "the half" to "half"

13. L187, change "polar-orbit" to "polar orbit"

14. L189, change "IASING" to "IASI-NG"

15. L229, change to "unavailability of satellite data or invalid retrievals"

16. L240, 290, change to "8 July 2010"

17. L262, change to "minimize the differences in . . ."

18. L264, remove "results"

19. L290, add "(blue)" after by "real AVK"

20. L314, change to "tackled"

21. L332, change to ", respectively" and "We then"

22. L391, change "Values of total DOF (0-60 km) . . ."

23. L511, change to "screens out"

24. L548-549, this sentence "is captured by IASI-NG+UVNS that is . . ." does not read well and needs to be rephrased.

25. L556-557, change to "is analysed", otherwise it is not a complete sentence

26. L584, change to "Same data as"

27. L602, change to "be considered"

28. L617, change to "constraint" as "constrain" is a verb

29. L618, change "relies a on" to "relies on the" or "relates to the"

---

## Author Comment (AC2) · 24 Feb 2017

We thank referee#1 and referee#2 for their questions and suggestions. Here, we provide a description of our answers, corrections and modifications to the manuscript. Apart from trivia and minor changes, corrected as well, the new text added in the manuscript is reported in red.

Referee#1

**The authors should explain how the following changes from GOME-2 to UVNS would affect the ozone data, e.g. the increase of retrieval throughput of near surface ozone due to the lower frequency of cloud contamination within field of view of satellite observation. "UVNS will have a higher signal-to-noise ratio (SNR) than GOME-2, a much finer horizontal resolution but a factor 2 coarser spectral resolution."**

We agree that it is useful to explain the gain expected by each instrumental improvement. We add the following text:

A finer resolution will increase the number of surface ozone retrievals, decreasing the number of cloudy pixels with CLF > 0.3. A finer (coarser) spectral resolution will increase (decrease) the retrieval vertical sensitivity to ozone. A higher SNR will improve the quality of the retrieval, leading to a higher vertical sensitivity and a smaller error. For better exploiting IASI-NG+UVNS, we have designed a constrain matrix accounting for the capability of the new sensors. As done by **Cuesta et al. (2013)**, we have adjusted the constraints to keep a similar retrieval error to IASI+GOME-2 and enhance the sensitivity between the surface and 3 km.

**The authors should add the actual spectral resolution used in the simulation/retrievals in the list of differences in nominal specifications since the spectral resolution is one of key parameters that determine the sensitivity of observation. The parameter of spectral sampling alone is not enough for this purpose.**

The spectral resolution is already present in Table 2. However, we agree that it should be placed next to the spectral sampling to be clearly visible. We re-ordered the table.

**Please add in maps of a priori ozone field and estimated uncertainty of near surface ozone to help in showing the performance of joint IASI/GOME2 retrievals catching the spatial distribution of near surface ozone.**

We do agree with referee#1 that maps of a-priori values and surface ozone estimated uncertainties would be useful to show the performances of IASI-GOME2 retrieval algorithm, with respect to single band retrieval methods. However, the «philosophy» of this article has been (from the

beginning) to show the improvement of using IASI-NG+UVNS method with respect to IASI+GOME-2 and not to « validate » and corroborate the robustness of IASI+GOME-2 retrieval algorithm and the gain with respect to single-band methods. Also because of the large quantity of images already present in the text (48), if referee and editor agree, we would like to continue to be coherent with this «philosophy» and leave the whole detailed analysis of IASI+GOME-2 product quality to the original paper by Cuesta et al. (2013).

**P 1 line 21: lowermost troposphere > lower most troposphere**

Changed

**P 1 line 28: to observer > to observe**

Changed

**P 2 line 42: Should "the surface-2km" change to "the surface-3km" since elsewhere in the manuscript always refer to "the surface-3km"?**

In this case, we do refer to surface-2km ozone.

**P 4 line 134: "(ozone concentration, skin temperature, temperature profile)" > "(ozone concentration and temperature profiles, skin temperature)"**

Changed

P 7 line 224: "AKV" > "AVK"

Changed.

**P 9 line 279 to 282: "The positive difference of 0.74 DU (7%) that remains between PR\*realAVK and real IASI+GOME-2 might be linked to systematic explained underestimation of cloud fraction. overestimation of ozone production in MOCAGE model, coming from the systematic underestimation of cloud fraction." The sentence is not clear and there is an extra "." in front of overestimation. Please consider to revise it, e.g., "The positive difference of 0.74 DU (7%) that remains between PR\*realAVK and real IASI+GOME-2 might be due to systematic underestimation of cloud fraction since the systematic underestimation of cloud fraction could lead to overestimation of ozone production in MOCAGE model, coming from the systematic underestimation of cloud fraction (reference therein).". Please add the reference(s) on the relationship between cloud fraction and ozone production, if there is. Or,**

**perform a brief sensitivity study on the impacts of cloud fraction on ozone production.**

We erase the sentence. We agree with referee#2 that there are many factor that can explain such a small positive difference. It is useless just to mention cloud fraction. However, it is a well-established knowledge that an increase solar radiation increases the photolysis and enhances the production of tropospheric ozone (e.g, Seinfeld and Pandis, Atmospheric Physics and Chemistry, 2006).

**P 10 line 294: "smoothed by real AVK" > "smoothed by real AVK (blue)"**

Changed.

**P 20 Figure 9 and Page 21 Figure 10: Besides showing the color bar with unit of mol/µm3, authors should consider add color bar of ppbv for the convenience of many readers in air quality community.**

Referee #2 also suggested this change. However, we already considered this change but the result is not convenient. Using ppbv, the strong vertical gradient of atmospheric density does not allow to see small variations in ozone concentration. It is necessary to multiply ppm * air_density * 10^-6 (i.e., molecules/cm³) to appreciate small vertical gradient of ozone.

**P 21 line 541: "9E" > "9°E"**

Changed.

Referee#2

**The abstract seems to be too long with a long introduction. Please make it concise and only highlight the important points.**

We did not made major text changes but we erased several sentences in the first part of the abstract and all the references. Now the abstract is much shorter and concise.

**The abstract typically should not have citations.**

Changed.

**Also, please define acronyms at their first occurrences such as IASI, IASI-NG, GOME-2, EPS-NG, UVNS, a.s.l., in the abstract, and also separately at their first occurrences in the text.**

Changed.

**L40, L43 and in the text, suggest adding "for signal" after "degree of freedom" as it could be used for signal or for noise.**

We added «for signal » the first time we speak about degrees freedom.

**L51, it seems to me that ADEOS is not currently working.**

Changed. ADEOS erased.

**L53, this also applies to global CTMs**

Changed.

**suggest adding "with adequate vertical sensitivity" after "observed" as UV retrievals can be sensitive to ozone down to below 2-3 km although with reduced sensitivity and retrieved at a much broader altitude range as shown in Liu et al. (2010)**

Added.

**8. L65, add "to enhance sensitivity to LMT ozone" after "spectral domains"**

Added.

**I would not call this "for the first time" as the study of Fu et al. (2013), published almost at the same time (slightly earlier), also showed similar capabilities (e.g., surface-700 hPa ozone)**

Changed. We erased «for the first time». We explained better in what main differences consists :

Due to the limited spatial coverage of TES (no across-track scanning is performed), this method was analysed in a profile-to-profile basis (**Fu et al., 2013**). **Cuesta et al. (2013)** developed a multispectral synergism of IASI (for TIR) and GOME-2 (for UV) spectra capable of observing from space the daily distribution of ozone plumes located below 3 km of altitude, defined here as the lowermost troposphere (LMT). This last multispectral approach (referred to as IASI+GOME-2 and used in the following of this paper) shows a particularly good accuracy (low mean bias near 1% and precision of 16%) and the capacity to observe the horizontal distribution of LMT ozone provided the scanning capacities of both IASI and GOME-2. IASI+GOME-2 allows the observation of LMT ozone due to an altitude of maximum sensitivity for the LMT peaking exceptionally low, on average at 2.2 km height over land (about 800 m below single-band methods), enhancing the degree of freedom (for signal) in the LMT by about 40% with respect to single-band retrievals.

**suggest rephrasing to "For practical consideration of computation, OSSEs typically approximate the pseudo-observations, : : :"**

Changed.

**The vertical resolution is not a property of radiation transfer models, but the radiative transfer calculations. I suggest rephrasing it to "All the profiles are sampled to 1-km layers in the radiative transfer calculations"**

Changed

**in Table 1, what is the SNR between 306-325 nm?**

The SNR is not reported in the Table because this band window is simply not used in the retrieval algorithm (interface between channel 1 and channel 2, large uncertainties in slit functions).

**the sentence "At given altitude (defined by the matrix column), each row shows " is confusing**

**and should be rephrased, as both column and row refer to a given altitude and the sentence reads like, "For the given column, each row**

Changed

**In section 4, it would be very useful to mention other retrieval variables (some of them are interfering) in addition to ozone even it is mentioned in Cuesta et al. (2013).**

We added:

The ozone profile (volume mixing ratio) is obtained by inverting the measurement vector (built up by merging together IASI TIR atmospheric radiances with GOME-2 UV earth reflectances) and jointly adjusting the water vapour profiles, offsets for each TIR micro-window, wavelength shifts for the UV radiance and irradiance spectra, multiplicative factors of the ring spectrum, surface albedo multiplicative factors and a factor for cloud fraction used in the UV forward calculations.

**the sentence is not accurate as the cloud can enhance ozone sensitivity above clouds, suggest rephrasing to "IASI+GOME-2 retrieval sensitivity to LMT ozone is significantly reduced when CLF > 0.3"**

Changed.

**I don't see the need to understand the small mean IASI+GOME-2 and MOCAGE differences as one is based on retrievals (not real data), and the other is based on assimilation of different ozone products, and too many important factors in retrievals, model simulations/assimilations can cause their differences.**

We agree. We erased the following sentence: "The positive difference of 0.74 DU (7%) that remains between PR*realAVK and real IASI+GOME-2 might be linked to systematic explained underestimation of cloud fraction. overestimation of ozone production in MOCAGE model, coming from the systematic underestimation of cloud fraction."

**P 17. L363, it is not clear about what you mean by "with a similar gap of 0.8 . . ."**

We erased this last part of the sentence.

**Table 1 & 2: captions typically are put above tables. Please check the requirements. Table 2**

**caption, it is good to describe what is R, σ-bias, and RMS in a few extra words. The gains for different columns are likely calculated inconsistently, please consistently use the IASI-GOME-2 as the reference point. The gain for the last one should be (1.55-1.77)/1.77=-12.4%**

Changed with corrected values. We made a mistake comparing data with the results of a specific day instead of averages values reported in Table. We apologise for this error and we really thank referee#2 for its attention and care in reviewing the article.

**In the 3rd row of Figure 5, there are no significant differences in the linear regression slope between black and red. Typically, you want to show how well your pseudo observations can capture the pseudo-reality or pseudo-reality with AVK (typically in validation). I think that it is better to show the PO (y-axis) vs (PR or PR*AVK). The slope for PO vs PR*AVK should be much closer to 1. This is also the case for Figure 12. Also in the 4th row, the standard deviation between PO and PO*AVK is much larger for the IASI-NG+UVNS (1.47 DU) than that of IASI+GOME2 (0.74), suggesting that the retrieval precision is worse for IASI-NG+UVNS. It is good to mention this and give an explanation.**

Changed. We agree that this information is more interesting and we changed the images as suggested by the referee. We report here the new figures in.

[Figure]

Figure 5

[Figure]

Figure 12

We also added new text

In chapter 3:

The constraint of the retrieval algorithm allows to define which parameter, between vertical sensitivity and retrieval error, is more improved. For better exploiting IASI-NG+UVNS, we have designed a constrain matrix accounting for the capability of the new sensors. As done by **Cuesta et al. (2013)**, we have adjusted the constraints to keep a similar retrieval error to IASI+GOME-2 and enhance the sensitivity between the surface and 3 km.

In Chapter 6.2 :

On the other hand, the correlation coefficient between PO and PR*AVK (red) decreases from 0.91 to 0.79 (-13%), as well as the slope of the linear fit slope (from 1 to 0.85, -15%), sing of a larger measurement error for IASI-NG+UVNS than for IASI+GOME-2 (quantified afterwards).

and

The total retrieval error, quantified by the standard deviation of the mean bias, does not vary significantly (from 1.76 to 1.72, -2%) because of the net balance between the decreasing smoothing error (from 1.59 to 1.28, -19%) and the increasing measurements error (from 0.74 to 1.47, +92%). At the same time, the root mean square (RMS) decreases from 1.78 to 1.63 (-8%). These results indicate an overall higher quality of the retrieval (smaller RMS) when using EPS-SG sensor instead of MetOp instrumentation, with a global gain in retrieval accuracy (smaller bias) and an almost constant retrieval precision (slightly smaller σ-bias).

and the RMS has been added to figure 6.

[Figure]

and we also added

The RMS of the PO-PR difference distribution, which is an estimate of the overall retrieval quality combining together accuracy and precision, decreases monotonically from IASI to IASI-NG+UVNS. For what concerns the bias and the σ-bias, results are consistent with the design of the constrain matrix of IASI-NG+UVNS which is less constrained than in IASI, IASI+GOME-2 and IASI-NG algorithms in order to increase the vertical sensitivity to near-surface ozone. This particular choice leads to a higher accuracy and a smaller smoothing error (lower bias) for IASI-NG+UVNS than all other retrieval methods, but also to a larger measurement error that enhances the total retrieval error (σ-bias) of IASI-NG+UVNS with respect to IASI-NG.

In : Chapter 7 (Conclusions)

The IASI-NG+UVNS retrieval algorithm parametrization has been optimized to increase at most the sensitivity in the lowest layers of the atmosphere (as close as possible to human biosphere), rather than decreasing the total retrieval error. Hence, the major gain of using the multispectral synergism of EPS-SG sensors with our retrieval approach concerns DOF-3km and Hmax-3km.

**21. Figure 9, I would like to suggest the same thing as the first reviewer, i.e., plotting the data in more conventional units of ppbv.**

Referee #1 also suggested this change. However, we already considered this change but the result is not convenient. Using ppbv, the strong vertical gradient of atmospheric density does not allow to see small variations in ozone concentration. It is necessary to multiply ppm * air_density * 10^-6 (i.e., molecules/cm³) to appreciate small vertical gradient of ozone.

**The synthetic retrievals show the potential of synergism of combining IASI-NG and GOME-2 for enhancing LMT retrievals. However, real retrievals might not be able to realize all the potentials due to mismatch of IASI-NG and UVNS and the retrieval challenges in TIR or in UV or the combination. Take the UV for example, the signal to noise ratio is assumed to be**

**1000. In practice, it might be difficult to fit the data well enough to this level. Also there can be strong correlation between surface albedo parameters and ozone in the lower troposphere, reducing the potential such enhancement to LMT ozone retrievals. It might be useful to discuss/mention some of the implementation challenges or future work in the conclusion.**
We agree. We added in the conclusion:

It is worth noting that additional challenge will be encountered for real retrievals using high SNR spectra from IASI-NG and UVNS. Indeed, as noise will be lower for such measurements, new sources of errors will be significant (i.e., surface properties accuracy, trace gasses variability). Such additional errors are not considered here and might degrade the capability of the retrieval. Results presented here are then an upper limit of the capability expected for IASI-NG+UVNS multispectral synergism to probe ozone pollution.